# Modulation of Immune Response and Cecal Microbiota by Dietary Fenugreek Seeds in Broilers

**DOI:** 10.3390/vetsci11020057

**Published:** 2024-01-28

**Authors:** Deependra Paneru, Guillermo Tellez-Isaias, Walter G. Bottje, Emmanuel Asiamah, Ahmed A. A. Abdel-Wareth, Md Salahuddin, Jayant Lohakare

**Affiliations:** 1Department of Poultry Science, University of Georgia, Athens, GA 30602, USA; dpaneru@uga.edu; 2Center of Excellence in Poultry Science, Division of Agriculture, University of Arkansas, Fayetteville, AR 72701, USA; gtellez@uark.edu (G.T.-I.); wbottje@uark.edu (W.G.B.); 3Department of Agriculture, University of Arkansas at Pine Bluff, Pine Bluff, AR 71601, USA; asiamahe@uapb.edu; 4Department of Animal and Poultry Production, Faculty of Agriculture, South Valley University, Qena 83523, Egypt; aahabdelwareth@pvamu.edu; 5Poultry Center, Cooperative Agricultural Research Center, Prairie View A&M University, Prairie View, TX 77446, USA; mdsalahuddin@pvamu.edu

**Keywords:** fenugreek, microbiota, immune response, gene expressions

## Abstract

**Simple Summary:**

Fenugreek seeds, a natural source of bioactive compounds, were tested for their effects on the immune system and gut bacteria of broiler chickens. Broilers were fed three diets: control, low fenugreek seed powder (5 g/kg), and high fenugreek seed powder (10 g/kg). The results showed that fenugreek seeds downregulated genes related to inflammation and antimicrobial defense, suggesting reduced inflammation, and improved immune response. Fenugreek increased the “good” Firmicutes bacteria and decreased the “bad” Actinobacteriota, Gemmatimonadota, and Verrucomicrobiota. It also boosted beneficial *Alistipes*, *Bacteriodes*, and *Prevotellaceae* bacteria. In conclusion, fenugreek seeds positively impact broiler chickens’ immune systems and gut bacteria, possibly by influencing how these two systems interact. This suggests the potential for fenugreek as a natural health promoter in poultry production.

**Abstract:**

Fenugreek seeds (FSs) are a natural source of bioactive compounds that may modulate the immune system and gut microbiota in broilers. This study examined the effects of dietary fenugreek seed powder on immune-related gene expression and cecal microbiota composition in broilers. A total of 144 broiler chickens were randomly allocated to three dietary groups, CON (0 g/kg FS, FS5 (5 g/kg FS) and FS10 (10 g/kg FS), each with 6 replicates of 8 birds. Ileum tissues and cecal contents were collected on day 42 for the mRNA expression of inflammation and antimicrobial defense-related genes and cecal microbiome diversity, respectively. The results indicated that fenugreek seeds downregulated mRNA-level inflammation and antimicrobial defense-related genes: IL6, IL8L2, CASP6, PTGS2, IRF7, AvBD9, AvBD10, and AvBD11. Moreover, fenugreek seeds altered the cecal microbial community by increasing the population of Firmicutes and decreasing the population of Actinobacteriota, Gemmatimonadota and Verrucomicrobiota at the phylum level and increasing *Alistipes*, *Bacteriodes* and *Prevotellaceae* at the genera level. These findings suggest that fenugreek seeds have a positive impact on the immunological profile and microbiome of broiler chickens, possibly through the interplay of the immune system and the gut microbiome.

## 1. Introduction

The poultry industry contributes to human food security by supplying essential nutrients and converting a variety of agricultural byproducts into edible meat and eggs [1,2]. Meeting the demands of a growing population requires optimizing poultry production. While antibiotic growth promoters (AGPs) have historically been used to improve growth rates, concerns about their impact on antibiotic resistance and the gut microbiome have led to increasing restrictions on their use [3,4,5]. Growing concerns about antibiotic resistance and potential human health consequences prompted food and health regulators to restrict the use of AGPs in poultry feed. This shift necessitates the development and adoption of effective non-antibiotic alternatives to maintain or improve poultry health and productivity, contributing to a more sustainable and responsible food system [6].

Fenugreek seeds (FSs), rich in phytochemicals, showcase diverse beneficial properties as an alternative to AGPs, impacting blood sugar, cholesterol, immunity, digestion and inflammation [7,8,9,10,11]. The major bioactive constituents of FS include saponins, alkaloids, flavonoids, steroidal saponins and dietary fiber [12]. Several studies have demonstrated that supplementing broiler diets with FS leads to improved growth outcomes due to the presence of essential fatty acids and high-quality proteins in FS, which provide essential nutrients for growth and development [13,14]. The gut microbiota interacts with the mucosal immune system and influences its development, function, and regulation [15]. Avian leukocytic cytokines are produced by immune cells and are important regulators of the immune response [16]. Interleukin-6 is a multifunctional cytokine that plays a major role in regulating immune responses, acute phase reactions, and hematopoiesis [17]. Phytogenics enhance gut immunity by altering the gut microbiota, lowering oxidative stress, and regulating cytokine expression [18]. Pytobiotics can be catabolized by the intestinal microbiota into phenolic acids and other metabolites that aid in lowering inflammation and oxidative stress in the intestines of broilers [19,20]. Additionally, it has been discovered that in broilers, phytogens help maintain intestinal integrity [21]. Bruce-Kellere et al. [22] revealed that the inclusion of 2% fenugreek seeds in mice showed a significant and robust effect on gut microbiota, with changes in alpha and beta diversity and taxonomic redistribution under both control and high-fat diet conditions. The growth performance and nutrient digestibility of rabbits were enhanced in our prior study by adding 15 g/kg of FS and a combination of probiotics to their diet [23]. Moreover, our previous research has explored the potential of fenugreek seeds to influence broiler growth, blood profiles, and gut health. Dietary fenugreek seeds at 0, 2.5, 5 and 10 g/kg linearly decreased the body weight gain of the birds during the starter phase, but this effect was only temporary and reversed during the finisher phase (22–42 days) [24].

While FS has shown promise in supporting broiler health, the lack of comprehensive research on its impact on immune-related gene expression and the cecal microbiome leaves a significant gap in understanding its potential benefits. Therefore, the objectives of this study are to assess the effect of fenugreek seeds on the immune-related gene expression and microbiome in male broilers.

## 2. Materials and Methods

### 2.1. Birds, Diets, and Management

This study employed previously described methods for animal husbandry, diet formulation, and the experimental design [24]. Briefly, one-day-old, male broiler chicks (Ross 708, Hot Springs, AR, USA, *n* = 144) were randomly allocated into 18 floor pens with fresh pine wood shavings. Each pen was equipped with a separate feeder and drinker (Tractor Supply Co., Pine Bluff, AR, USA). The 18 pens were assigned to 3 dietary treatment groups (6 pens/treatment; 8 birds/pen) in a completely randomized design. The dietary treatments consisted of three groups: a control group fed with basal diet without fenugreek seeds (CON), a basal diet with 5 g/kg fenugreek seeds (FS5), and a basal diet with 10 g/kg fenugreek seeds (FS10) during the starter (0 to 21 days) and grower–finisher (22 to 42 days) periods. The formulations for the starter and grower–finisher diets were published previously [24]. We added fenugreek seeds (Deep Foods Inc., Union City, NJ, USA) to the starter and finisher diets after grinding them to medium texture in a grinding grain mill (Thomas Scientific, Swedesboro, NJ, USA) with a 1 mm sieve. The active compounds in the extract of fenugreek seeds powder were analyzed in our previous study by Abdel-Wareth et al. [23], using gas chromatography–mass spectrometry.

Throughout the experimental period, the birds were continuously provided with ad libitum feed and water. The housing temperature was set at 34 °C on day 0 and gradually reduced to 25 °C by day 42. On day 42, two birds per pen were selected that represented the pen and euthanized via decapitation for the collection of ileum tissue from the middle portion of the ileum and cecal contents from both ceca. All tissue and cecal contents were snap-frozen in liquid nitrogen and stored at −80 °C until further analysis.

### 2.2. RNA Isolation and RT-qPCR

Total RNA was isolated from ileum tissues using the PureLink RNA Micro Kit (Invitrogen, Carlsbad, CA, USA) and on-column digestion with DNase following the manufacturer’s recommendations. The working surfaces were decontaminated with RNaseZapTM (Invitrogen, Carlsbad, CA, USA) solution to prevent RNase contamination. RNA concentration (ng/μL) and purity (A260/A280) were assessed using a NanoDrop 8000 Spectrophotometer (ThermoScientific, Waltham, MA, USA). RNA quality was verified with gel electrophoresis using a 2% agarose E-Gel (Invitrogen, Carlsbad, CA, USA). A pure RNA sample (2 μg) was then reverse-transcribed to synthesize cDNA using the Maxima cDNA Synthesis Kit (ThermoFisher Scientific, Waltham, MA, USA) following the recommendations of the supplier.

The oligonucleotide primers specific for chicken Avian β-defensin 9 (AvBD9), Avian β-defensin 10 (AvBD10), Avian β-defensin 11 (AvBD11), Interleukin 6 (IL6), Interleukin 8-like 2 (IL8L2), Caspase 6 (CASP6), Prostaglandin-endoperoxide synthase 2 (PTGS2), Interferon regulatory factor 7 (IRF7), and β-actin (ACTB) were designed based on the NCBI gene database using Primer3Plus version 3.2.0 software (Table 1) and purchased from Invitrogen, CA, USA. ACTB was used as a housekeeping gene.

The QuantStudio 6 Pro (Applied Biosystems, Waltham, MA, USA) was used for the real-time polymerase chain reaction (RT-PCR) assay. In a 20 μL reaction volume, 400 nM of both forward and reverse specific primers and SYBR Green Master Mix (Applied Biosys-tems, Waltham, MA, USA) were used to amplify 10 ng of cDNA. The initial RT-PCR protocol involved a 2 min incubation at 50 °C followed by a 10 min denaturation step at 95 °C. Subsequently, 40 cycles of amplification were performed, each consisting of a 15 s denaturation step at 95 °C and a 1 min annealing/extension step at 58 °C. Melting curve analysis was then conducted to confirm target gene amplification. Relative target gene expression was then assessed using the 2^−ΔΔCt^ method, normalizing to housekeeping gene β-actin expression.

### 2.3. Microbial DNA Extraction and 16S rRNA Sequencing

Six cecal content samples were collected from each replicate group of birds. To preserve the genetic material within, the samples were immediately stored at −80 °C. DNA extraction was then performed using the QIAamp DNA Stool Mini Kit from Qiagen (Hilden, Germany). The extracted DNA was subsequently quantified and assessed for purity using a NanoDrop One spectrophotometer from Thermo Fisher Scientific (Madison, WI, USA). The amplification of the V3-V4 regions of the 16S rRNA gene was performed using the primers (5′-GTGCCAGCMGCCGCGGTAA-3′) and (5′-GGACTACHVGGGTWTCTAAT-3′), with PCR reactions conducted using Phusion^®^ High-Fidelity PCR Master Mix from New England Biolabs Inc. (Ipswich, MA, USA). Following amplification, the generated DNA fragments, known as amplicons, were assessed for size and purity using agarose gel electrophoresis. A 2% agarose gel was prepared, stained with a DNA dye, and loaded with the PCR products alongside a size marker. Successful amplification was confirmed by the presence of distinct bands within the expected size range of 400–450 bp. These bands represent the amplified V3–V4 region of the 16S rRNA gene. Amplicons within this size range were then purified using the Qiagen Gel Extraction Kit (Qiagen, Germany). The purified amplicons were subsequently prepared for sequencing using the NEBNext^®^ Ultra™ DNA Library Prep Kit for Illumina, San Diego, CA, USA.

Subsequent bioinformatics analysis involved demultiplexing paired end reads via FLASH (V1.2.7), quality filtering with QIIME (V1.7.0), and chimera detection using UCHIME algorithm against the SILVA reference database. Effective tags were clustered into OTUs at 97% similarity with Uparse software (v7.0.1090), and representative sequences were annotated with QIIME against the SILVA SSUrRNA database. Diversity metrics, both alpha and beta, were calculated using QIIME and subsequently visualized using R software (Version 2.15.3).

Alpha diversity is applied in analyzing the complexity of biodiversity for a sample through the Shannon, Simpson, ACE, and Chao indices. To evaluate differences in species complexity between samples, we employed beta diversity analysis. Weighted and unweighted UniFrac metrics were calculated using QIIME software (version 1.7.0) to provide complementary perspectives on community structure. Prior to cluster analysis, principal component analysis (PCA) was applied to reduce the dimensionality of original data and identify key patterns. This dimensionality reduction was performed using the FactoMineR package in R. The resultant PCA plot, often visualized as a principal coordinates analysis (PCoA) plot, provided a visual representation of beta diversity. Following PCA, cluster analysis was performed to identify groups of samples with similar microbial profiles.

### 2.4. Statistical Analysis

The experimental unit for all analyses was the replicate pen. One-way ANOVA (SPSS V28.0, IBM Corp., Armonk, NY, USA) was used to analyze the data. The treatment groups were compared using Duncan’s multiple range test. A significance level of *p*-value < 0.05 was used, and *p*-values between 0.05 and 0.10 indicated a trend. The results are presented as the means and their pooled standard error. Additional statistical evaluations, such as Metastat, were performed in R software. The LEfSe analyses were carried out using the specialized LefSe version 1.1.2 software. All these indices in our samples were calculated with QIIME (Version 1.7.0). The *p*-values were computed through permutation testing, providing a measure of statistical significance. In parallel, q-values were calculated by employing the Benjamini and Hochberg false discovery rate method, which aids in controlling the rate of false positives in multiple hypothesis testing. For the identification of statistical significance among the three groups, linear discriminant analysis effect size (LEfSe) analysis was performed using the LEfSe software.

## 3. Results

### 3.1. Relative Expression of Immune-Related Genes

#### 3.1.1. Inflammation-Related Genes

The inclusion of fenugreek seeds linearly downregulated (*p* < 0.05) the expression of the IL6 gene compared to the control group (Figure 1A). Additionally, a significant reduction in IL8L2 gene expression was observed with fenugreek seed inclusion, with the FS5 group displaying the lowest expression (Figure 1B). Furthermore, dietary fenugreek seeds demonstrated a significant downregulation of Caspase-6 expression in the broiler ileum (Figure 1C). Moreover, the mRNA expression of PTGS2, a crucial marker of ferroptosis, exhibited a linear decrease with dietary fenugreek seed inclusion (Figure 1D).

#### 3.1.2. IRF7 and Avian Defensins

The dietary inclusion of fenugreek seeds significantly downregulated IRF7 expression, a key regulator of adaptive immune responses (Figure 2A). Examining the impact on ileal avian β-defensins (AvBD9, AvBD10, and AvBD11), a uniform and significant downregulation in their expression was observed across all dietary fenugreek concentrations compared to the control group (Figure 2B–D).

### 3.2. Composition and Biodiversity of the Cecal Microbiota

#### 3.2.1. Microbial Enrichments

The microbial community composition exhibited a consistent pattern by the comparative investigation of the bacterial phylum and genus distribution among three dietary groups: CON, FS5, and FS10. Bacteroidota (64 to 68%) and Firmicutes (19 to 26%) emerged as the most abundant phyla across all three groups. Dietary treatments affected the relative abundance of the microbiota at the phylum level (Figure 3A). Bacteriodota, Firmicutes and Cynobacteria were more abundant in the dietary fenugreek groups, whereas Actinobacteria, Acidobacteria, Gemmatimonadota, and Verrucomicrobiota were less abundant. The presence of other phyla such as Verucomicrobiota, Desulfobacterota, and Chloroflexi was observed to a lesser extent. These minor phyla, along with those categorized under ‘Others’, contributed to the diversity within the microbiota but did not dominate the profile. At the level of the genus, Alistipes was the most abundant genera across all three dietary treatment groups, with relatively higher abundance in the dietary fenugreek groups (Figure 3B). Barnesiella was lower in the fenugreek-supplemented group. Other genera including Bacteroides, Prevotellaceae_UCG-001, Clostridia_vadinBB60_group, Ruminococcus_torques_group, Parasutterella, Megamonas, Rikenellaceae, and Faecalibacterium contributed as major genera across the dietary groups.

#### 3.2.2. Alpha and Beta Diversity

In a comprehensive evaluation of microbial community diversity across three distinct groups, CON, FS5, and FS10, distinct patterns emerged from the alpha and beta diversity analyses (Figure 4 and Figure 5). The ACE and Chao index, which estimates the total species richness, pointed to a significantly higher median richness in the CON group, with a notable decrease in species richness in the FS5 and FS10 treatments. Despite this variation in species richness, the Shannon and Simpson indices, which assess the abundance and distributional evenness of species, respectively, revealed no significant differences in diversity among the groups, implying that the overall complexity of the community structure remains stable. Complementing these findings, our beta diversity analysis, which explores the differences in microbial community composition between the groups, highlighted distinct separations. Ordination methods such as NMDS and UniFrac metrics demonstrated that the microbial communities in treatment groups FS5 and FS10 are compositionally distinct from the control group. These results underscore the influence of treatments on community composition, despite the alpha diversity indices suggesting stability in diversity and evenness within the communities. Together, these alpha and beta diversity metrics provide a nuanced view of microbial ecosystem dynamics, revealing that the treatments affect community composition without significantly altering intra-community diversity.

#### 3.2.3. Differential Abundance Analysis Using Meta Stat

Meta Stat analysis indicated significant differences in the microbial composition between the CON and the treated FS5 and FS10 groups (Figure 6). Significantly, group FS5 had a higher abundance of Firmicutes, which are associated with improved gastrointestinal functionality, potentially aiding in the digestion and absorption of nutrients. In the same group, there was also a significant increase in Cyanobacteria and Desulfobacteria compared to both the control and FS10 groups. In contrast, the populations of Actinobacteriota, Acidobacteriota, Gemmatimonadota, Chloroflexi, Myxococcota, Entotheonellaeota, and Nitrospirota were significantly reduced in the dietary fenugreek seed groups compared to the control group.

#### 3.2.4. Characterization of Unique Microbiota

Using LEfSe to compare the microbial compositions of three groups revealed the presence of separate bacterial communities that were exclusive to each group (Figure 7). The enrichment of phylum Actinobacteriota and class Alphaproteobacteria in the control group was statistically significant, suggesting that these taxa are representative of the untreated baseline condition. While FS5 treatment significantly increased the number of taxa belonging to the phylum Firmicutes, class Clostridia and family Clostridia_vandinBB60_group, thereby distinguishing its microbial profile from both FS10 and the control. Treatment FS10 was notably associated with the family Lachnospiraceae, order Lachnospirales, family Prevotellaceae and genus Provotella_UCG_001 suggesting a distinct microbial shift induced by this treatment. Collectively, these findings demonstrate that each treatment exerts a unique influence on the microbial ecosystem when compared to the control.

## 4. Discussion

The symbiotic interaction between cecal microbiota and the host can influence the host’s immunological functions, as well as their nutrition, physiology, and metabolism. Commensal bacteria are important immune response modulators and a constant source of stimulation for an immune system [15]. Advancements in high-throughput omics technologies have broadened our understanding of the importance of commensal bacteria in the development of the chicken immune system. Poultry gut microbiome development can be influenced by a number of variables, one of which is feed. The innate immunity of broilers is influenced by genes that are involved in the production of interleukins, β-defensins, caspase, prostaglandin-endoperoxide synthase-2, and interferon regulatory factor 7 [25]. We tested eight different genes related to innate immunity and found significant downregulation (*p* < 0.05) of all the genes, with the inclusion of different dietary fenugreek concentrations. Epithelial cells respond to injury or infection by secreting cytokines, which are regulatory signals that control epithelial cell proliferation, differentiation, and function during inflammation [16]. Interleukin-6 (IL-6) is a versatile cytokine that plays a crucial role in the modulation of immune responses, acute phase reactions, and hematopoiesis [17]. IL-6 is produced by T cells and macrophages, serving dual roles as both a pro-inflammatory agent linked to the synthesis of acute-phase proteins and an anti-inflammatory cytokine. Notably, elevated IL-6 levels in chickens have been linked to infections caused by *Salmonella* and *Eimeria* [26,27]. Therefore, the downregulation of IL-6 mostly favors an anti-inflammatory response and shows that fenugreek seeds had an anti-inflammatory effect in the intestine. Avian β-defensins genes encode the cationic peptides in chickens and are found in many tissues, including the epithelium of the epidermis, gastrointestinal tract lining, respiratory tract and urogenital tracts of birds [28]. Beta-defensins play a crucial role in harmonizing innate and adaptive immune responses, alongside their direct antibacterial effects [29]. The avian beta-defensin 9 (AvBD9) plays a crucial role in the homeostasis of gastrointestinal microbiota and the intestinal immune system [30]. Some β-defensin genes are expressed all of the time in some tissues, and their expression can be upregulated in others in response to microbial infection or pro-inflammatory stimuli [30]. In other studies, AvBD10 was upregulated in response to *Salmonella* infection in the broiler chicken [31]. Evidence suggests that the pathogenic process initiated by a bacterial infection stimulates the expression of several genes in poultry, including those involved in the synthesis of avian β-defensins (AvBD9, AvBD10, and AvBD11) [25]. Our study indicated significant downregulation of AvBD9, AvBD10, and AvBD11 with the inclusion of dietary fenugreek seeds. Therefore, the downregulation of AvBD9, AvBD10, and AvBD11 indicates the absence of infection and shows that fenugreek seeds might reduce the infection caused by pathogens. Caspases regulate apoptosis, immune responses, and homeostasis. Caspase-6 (CASP6) mediates innate immunity and inflammasome activation in response to apoptosis caused by pathogen infection [32]. PTGS2 is responsible for the production of prostaglandins during inflammation [33]. Chicken IRF7 is an important regulator of type I interferon production. Interferons (IFNs) are critical factors in fighting viral infections, and they constitute the first line of animal and human defense during infection [34]. Studies have found significant upregulation of CASP6, PTGS2, and IRF7 with the pathological process initiated by bacterial or viral infection [25,34]. Our study indicated the significant downregulation of CASP6, PTGS2, and IRF7 with the inclusion of dietary fenugreek seeds. However, further studies are needed to evaluate the effect of FS on the host immune system under different stress conditions, such as infection or heat stress. Fenugreek seed extracts may have a beneficial impact on broiler immunity by regulating cytokine levels and stimulating the release of immunoglobulins [35]. In the current study, the microbial community composition exhibited a consistent pattern, as shown by the comparative investigation of the bacterial phylum distribution among groups CON, FS5, and FS10. The formation of intestinal integrity, control of appropriate immune responses, and competition with bacteria that are harmful or opportunistic are only a few of the functions of the gut microbiota that contribute to gut physiology and health [36].

The current study showed that the microbial community composition exhibited a consistent pattern, as shown by the comparative investigation of the bacterial phylum distribution among control groups, FS5, and FS10. Firmicutes and Bacteroidota emerged as the most abundant phyla across all groups, with Bacteroidota showing a slightly higher prevalence in group FS10. The effect of fenugreek on intestinal microbiota composition in broilers is still unknown due to the lack of studies on it. However, Bruce-Keller et al. [22] revealed that the inclusion of 2% fenugreek seed in mice showed a significant and robust effect on gut microbiota, with changes in alpha and beta diversity and taxonomic redistribution under both control and high-fat diet conditions. Gut morphology and integrity, as well as immunological responses, are impacted by the interactions between intestinal microbes and their host [37]. Phytogenic feed additives have gained growing significance in broiler nutrition because of their various positive effects on gut microbiota [38]. Phytogenic feed additives in broiler diets have been shown to reduce gut pathogenic bacteria like *E. coli* with the inclusion of thyme and star anise [39] and *Salmonella* with the inclusion of cinnamaldehyde and formic acid. Mountzouris et al. [40] reported an increase in beneficial members like *Lactobacillus* and *Bifidobacterium*, including a blend of essential oils from oregano, anise, and citrus in the broiler diet. There have also been reports of no effects on gut commensal bacteria [41,42] with different phytogenic feed additives. Paraskeuas and Mountzouris [38] concluded that the effect of phytogenic feed additives on gut microbiota composition varies with the composition of PFA, their inclusion level, farm hygiene, and the analytical approach used for gut health analysis.

In this study, fenugreek seeds had significant impacts on the immune system and the gut microbiota of chickens. Phytogenics enhance gut immunity by altering the gut microbiota, lowering oxidative stress, and regulating cytokine expression [18]. Phytobiotics can be metabolized by intestinal microbiota into metabolites, such as phenolic acids that aid in lowering inflammation and oxidative stress in the intestines of broilers [19,20]. Additionally, it has been discovered that in broilers, phytogenics control the immune system and maintain the gut’s integrity [20,21].

## 5. Conclusions

In conclusion, this study suggests that fenugreek seeds may modulate the expression of immune-related genes and influence the diversity of cecal microbiota in broiler chickens. However, further research is necessary to establish a causal relationship between fenugreek seed consumption and these observed effects. Additionally, future studies should investigate the potential benefits of fenugreek seeds in mitigating the impact of specific poultry pathogens, such as Coccidia or *Salmonella*, using controlled infection models.

## Figures and Tables

**Figure 1 vetsci-11-00057-f001:**
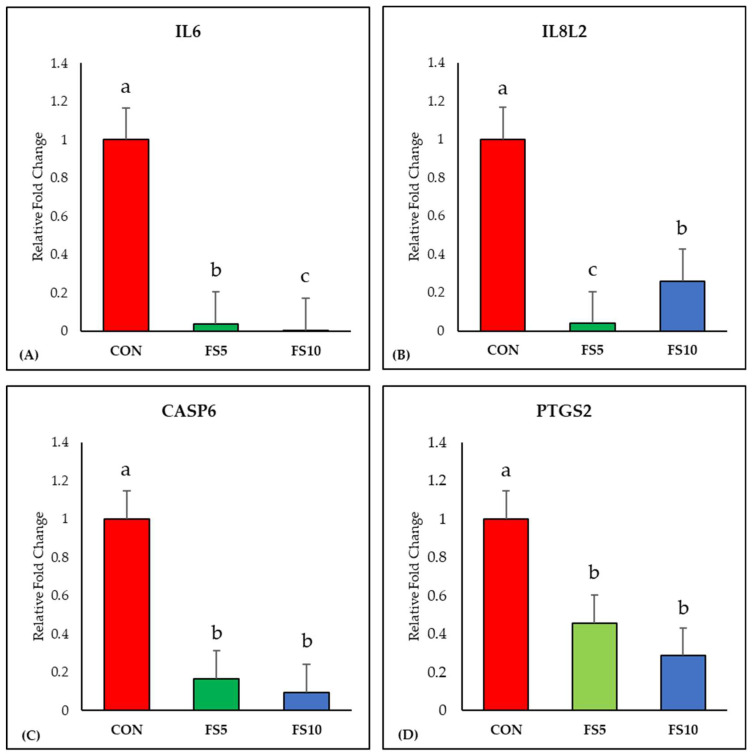
Relative mRNA expression of inflammation-related genes in the ileum of broilers fed with different dietary fenugreek concentrations. Genes: interleukin-6 (**A**), interleukin-8-like-2 (**B**), caspase 6 (**C**) and prostaglandin-endoperoxide synthase 2 (**D**). Abbreviations: CON = basal diet; FS5 = basal diet supplemented with 5 g/kg fenugreek seed; FS10 = basal diet supplemented with 10 g/kg fenugreek seeds. The letters a, b, and c in the bar plots represent significant differences (*p* < 0.05) between the treatment groups when they vary in the same bar plot.

**Figure 2 vetsci-11-00057-f002:**
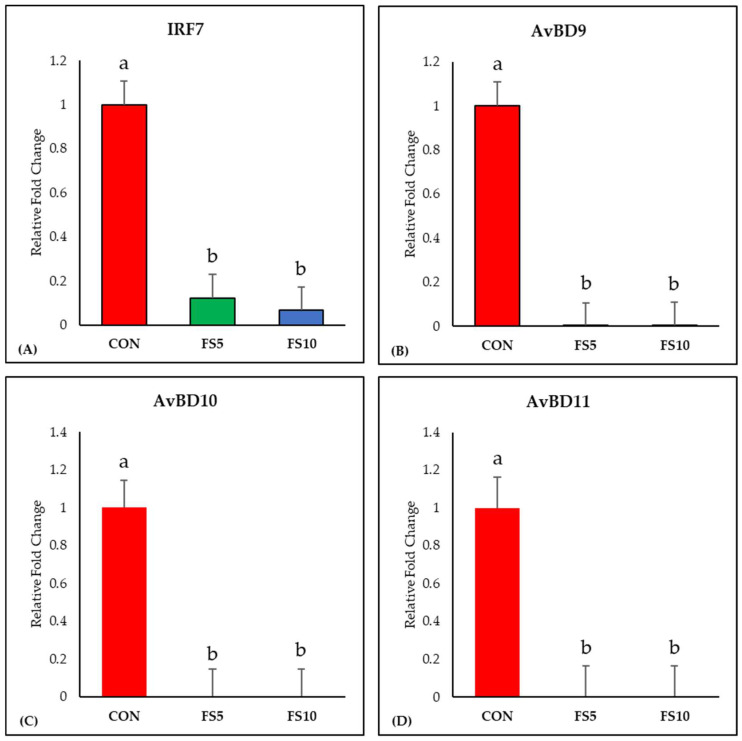
Relative mRNA expression of interferon regulatory factor 7 and antimicrobial defense-related genes in the ileum of broilers fed with different dietary fenugreek concentrations. Genes: Interferon Regulatory Factor 7 (**A**), Avian Beta Defensin 9 (**B**), Avian Beta Defensin 10 (**C**) and Avian Beta Defensin 11 (**D**). Abbreviations: CON = basal diet; FS5 = CON with 5 g/kg fenugreek seeds; FS10 = CON with 10 g/kg fenugreek seeds. The letters a and b in the bar plots represent significant differences (*p* < 0.05) between the treatment groups when they vary in the same bar plot.

**Figure 3 vetsci-11-00057-f003:**
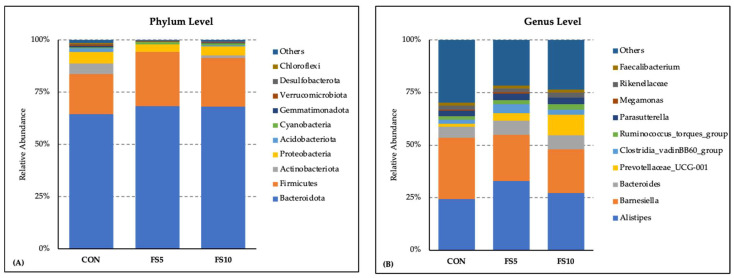
Composition of the cecal microbiota by genus and phylum level in the broilers fed with different dietary fenugreek concentrations. Bacterial phyla (**A**) and genera (**B**) relative abundance in three dietary groups: CON (basal diet); FS5 (CON with 5 g/kg fenugreek seed); FS10 (CON with 10 g/kg fenugreek seeds); n = 6/group.

**Figure 4 vetsci-11-00057-f004:**
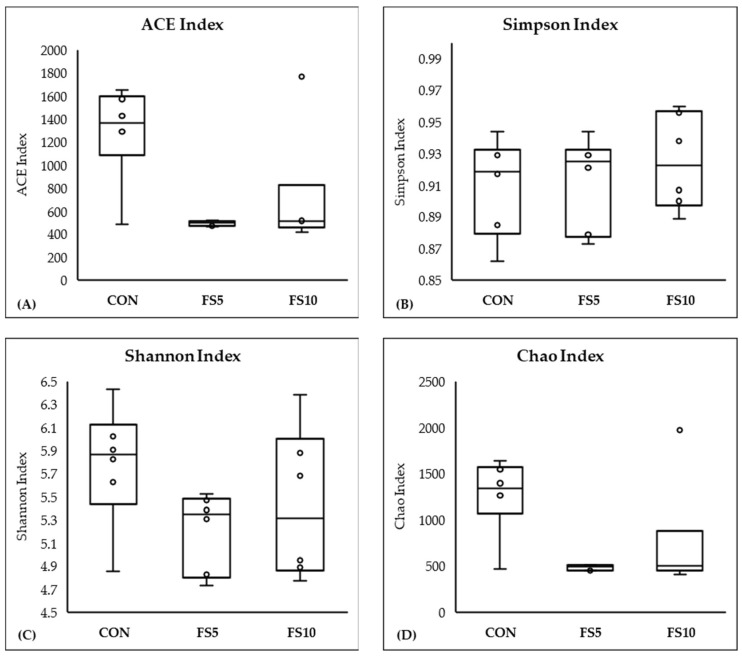
Microbial alpha diversity in the cecal contents of broilers fed with different dietary fenugreek concentrations. Alpha diversity indices: ACE index (**A**), Simpson index (**B**), Shannon index (**C**), and Chao index (**D**); n = 6/group. Abbreviations: CON = basal diet; FS5 = CON with 5 g/kg fenugreek seeds; FS10 = CON with 10 g/kg fenugreek seeds. Circles represent the data points for each treatment group.

**Figure 5 vetsci-11-00057-f005:**
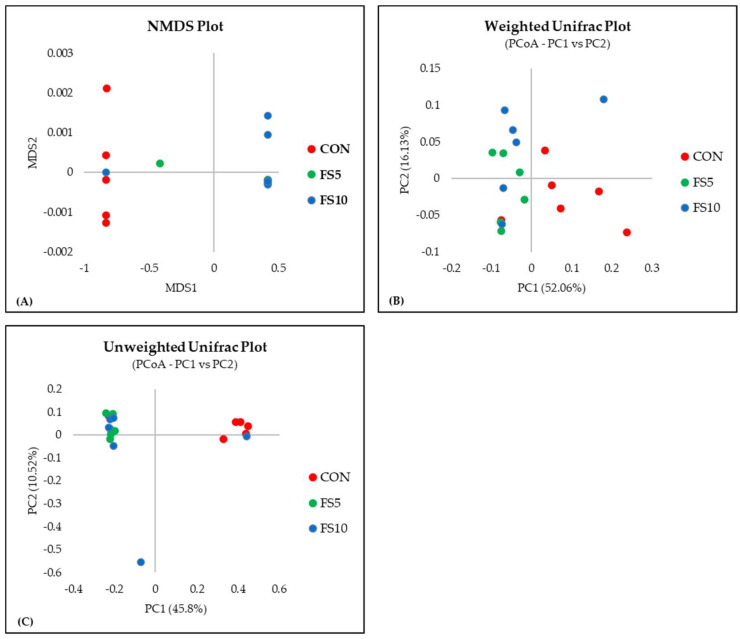
Microbial beta diversity in the cecal contents of broilers fed with different dietary fenugreek concentrations. Beta diversity indices: NMDS plot (**A**), weighted Unifrac plot (**B**), and Unweighted UniFrac Plot (**C**); n = 6/group. *p* < 0.05 indicated that the treatment effect was significantly different. Abbreviations: CON = basal diet; FS5 = CON with 5 g/kg fenugreek seeds; FS10 = CON with 10 g/kg fenugreek seeds.

**Figure 6 vetsci-11-00057-f006:**
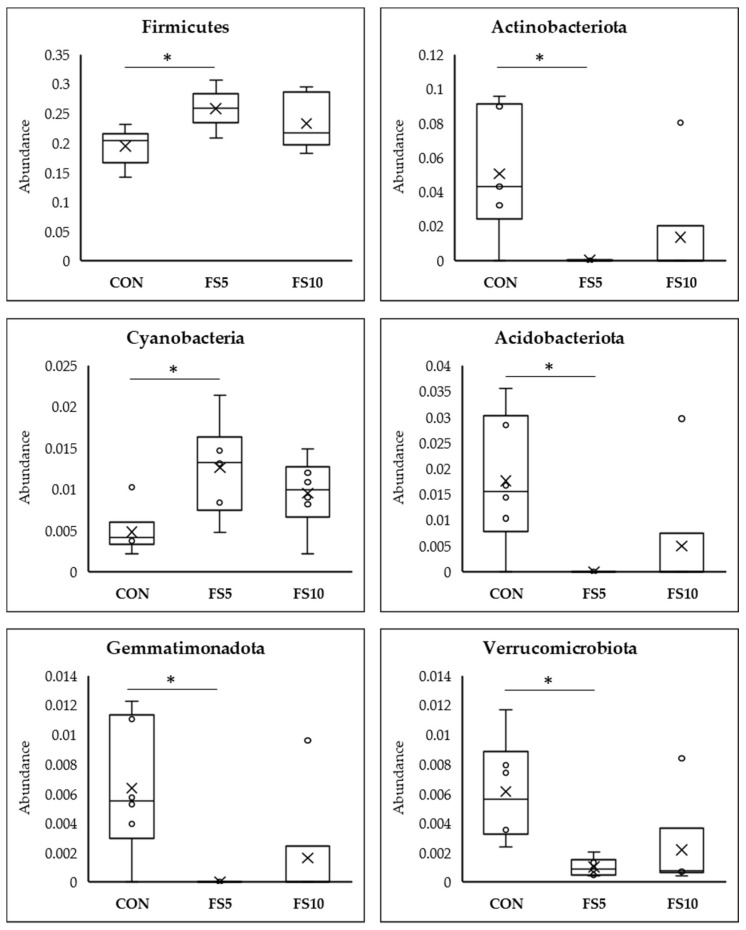
Top 12 bacterial phyla present in the cecum of broilers fed with different dietary fenugreek concentrations; n = 6/group. Abbreviations: CON = basal diet; FS5 = CON with 5 g/kg fenugreek seeds; FS10 = CON with 10 g/kg fenugreek seeds. Circle represents the data points and ‘X’ represents the mean value for each treatment group. Asterisk (*****) represents significant variation between the treatment groups (*p* < 0.05).

**Figure 7 vetsci-11-00057-f007:**
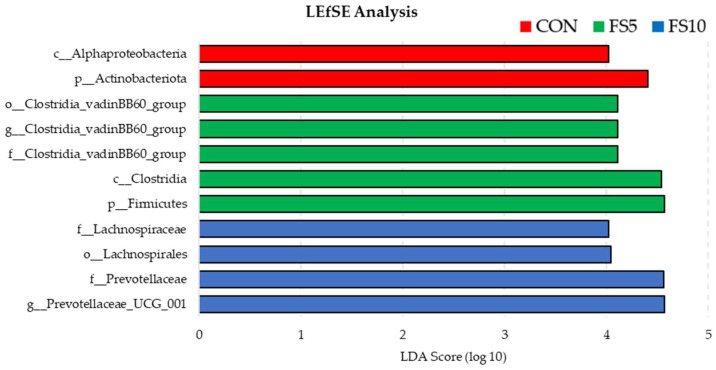
Bacterial community differences in the ceca of birds fed with different dietary fenugreek concentrations using linear discriminant analysis effect size (LefSE); n = 6/group. Abbreviations: CON = basal diet; FS5 = CON with 5 g/kg fenugreek seeds; FS10 = CON with 10 g/kg fenugreek seeds.

**Table 1 vetsci-11-00057-t001:** Specific oligonucleotide primers for real-time quantitative PCR.

Gene ^1^	Accession Number	Primer Sequence (5′ → 3′)
AvBD9	NM_001001611.2	F: GTCAGGCATCTTCACAGCTG
		R: GGCTAGGACTTCTCTGTGCA
AvBD10	NM_001001609.2	F: CACGTCCTGTTAGCACACTG
		R: AGCTGCATGAACCCAAAGTG
AvBD11	NM_001001779.1	F: CCCTCCTTCAGTTTCCCCTT
		R: CATCTGACTCACTGCTGCAC
IL6	NM_204628.1	F: TGGAAGAAGCATGGAGAGCA
		R: GCATCCGTTCCTATGTGCTG
IL8L2	NM_205498.1	F: CCGGATATGCAAACACTGGC
		R: AGAATTGAGCTGAGCCCTGT
CASP6	NM_204726.1	F: CGTGTTCAGTTGGACAGCAA
		R: GGAGGGTGCAAAACTGAAGG
PTGS2	NM_001167718.1	F: AAAGGGGCCAGTACTGTGTT
		R: TGCCCAGACTTGTCTTCCTT
IRF7	NM_205372.1	F: CACATGTTCATGCTGCTGGA
		R: CTGGAAGGAGAGCAGGTTGA
ACTB	NM_205518.1	F: ACTCTGTCTGGATTGGAGGC
		R: AAAGCCATGCCAATCTCGTC

^1^ AvBD9, avian β-defensin 9; AvBD10, avian β-defensin 10; AvBD11, avian β-defensin 11; IL6, interleukin 6; IL8L2, interleukin 8-like 2; CASP6, caspase 6; PTGS2, prostaglandin-endoperoxide synthase 2; IRF7, interferon regulatory factor 7; ACTB, β-actin.

## Data Availability

The datasets generated during the current study are available from the corresponding author upon reasonable request.

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
