# Peer review of "Modulation of Immune Response and Cecal Microbiota by Dietary Fenugreek Seeds in Broilers"

_vetsci, 2024, doi:10.3390/vetsci11020057_

Round 1
Reviewer 1 Report
Comments and Suggestions for Authors
This study was undertaken to investigate the impact of different concentrations of Fenugreek on the broilers' immune system and cecal microbiota composition. The results revealed that the inclusion of fenugreek in the diet at 5 g/kg (FS5) and 10 g/kg (FS10) downregulated various immune genes altered the gut microbiota.
Overall, multiple aspects of the manuscript require revision before it is deemed suitable for publication.
Abstract
· Remove the word “differentially” line 33.
Introduction
· Re-cast sentence on lines 58-60
· Sentences on Lines 64-65 and 67-69 portray the same point and appear similar. Please re-cast to make a better meaning.
Materials and Methods
· · Line 106 Why did the researchers choose to assess gene expression in the ileum rather than in other sections of the intestine and gut-associated lymphoid tissues, such as the cecal tonsil?
· What is the rational of measuring the gene expression of IL-6, IL-8, AvBD, PTGS2 among various immune system genes?
Results
· Please move the following lines to the discussion section (lines 182-183 starting from suggesting…….., 185-186 starting from indicating……….., 188-189 starting from emphasizing………., and 191-192 starting from suggesting)
· Line 186… is the dose-dependent relationship direct or inverse? Please specify (see Fig 1b). Secondly, do the FS5 (5g/kg) and FS10 (10g/kg) outcomes in Fig 1b, really substantiate a dose-dependent relationship?
· Revise the statement in Line 205…. replace it was observed that with “a uniform and significant downregulation in their expression was observed……………”
· Line 292: remove “and” from suggesting and…..
Discussion
· Revise lines 301-203 starting with the gut microbiota…
· Line 311 Avian Leukocytes cytokines: Is the production of cytokines exclusive to leukocytes among different cell types? …how about epithelial cells? Especially when you're measuring the gene expression in the ileum.
· Line 309… remove ‘s’ from ‘contribute’.
· Lines 318, 328, 374: italicize Salmonella
· Lines 329-335: is the downregulation of AvBD good for the host? It is well-known that AVBD has broad-spectrum antimicrobial activity against various Gram-negative bacteria and enhancing their expression is important for fighting infections. Also, does the downregulation of IRF-7 help the host fight viral infection? The observed downregulation of these genes requires clarification.
· Revise statement 346-348: the impact of fenugreek on the host immune system has not been examined in infected or heat-stressed birds in this study.
· The statement in lines 357-358 is awkward.
· Line 384: “In this investigation, the effects of fenugreek seeds on immune-related gene and cecal bacterial diversity were barely apparent” Is this true? How about the significant downregulation of the immune related genes.
· Segment 392-407, the authors' intended emphasis in this segment is unclear.
Conclusion
· Lines 410-411 and 413-414 are similar. The conclusion needs to be written to better reflect the outcomes of the study. Future directions should also be included.
Comments on the Quality of English LanguageSome awkward statements require revision.
Author Response
Response to Reviewer 1 Comments
|
|
We are grateful for your valuable comments and suggestions on our manuscript. We have carefully revised our manuscript following your comments and have made the corrections. We have also provided a point-by-point response to each comment in the following document. We have addressed all the suggestions in the revised manuscript. Authors’ responses are in red. We hope that our revised manuscript meets your expectations and is now suitable for publication.
|
|
Reviewer 1
- Abstract: Comment 1 Remove the word “differentially” line 33.
Response 1: Thank you for your suggestion. We have removed the word “differentially” from line 33.
- Introduction:
Comment 2: Re-cast sentence on lines 58-60
Response 2: Thank you for your suggestion. We have modified the sentence to say “The gut microbiota interacts with the mucosal immune system and influences its development, function, and regulation. It also provides a constant source of antigens that keep the immune system alert.”
Comment 3: Sentences on Lines 64-65 and 67-69 portray the same point and appear similar. Please re-cast to make a better meaning.
Response 3: We have rephrased the two sentences to not repeat the same point. Lines 64-64 and 67-69 have been modified to say that “Additionally, it has been discovered that in broilers, phytogenic helps to maintain the intestinal integrity.”
- Materials and Methods:
Comment 4: Line 106 Why did the researchers choose to assess gene expression in the ileum rather than in other sections of the intestine and gut-associated lymphoid tissues, such as the cecal tonsil?
Response 4: Thank you for your thoughtful question. The decision to focus on the ileum was made based on the specific objectives and hypotheses of our study. The ileum, being the terminal part of the small intestine, plays a crucial role in nutrient absorption and immune response regulation.
Comment 5: What is the rational of measuring the gene expression of IL-6, IL-8, AvBD, PTGS2 among various immune system genes?
Response 5: Thank you for your inquiry regarding the rationale behind measuring the gene expression of IL-6, IL-8, AvBD, and PTGS2 among various immune system genes in our study. The selection of these genes is based on their roles in immune response regulation, inflammation, and defense mechanisms.
- IL-6 is a pro-inflammatory cytokine that plays a crucial role in orchestrating immune responses and inflammation.
- IL-8 is a chemokine involved in the recruitment and activation of neutrophils, essential components of the immune system's first line of defense against infections.
- Avian β-defensins are antimicrobial peptides with broad-spectrum activity against bacteria and fungi. Their gene expression levels are indicative of the host's ability to mount an effective defense against pathogens.
- Results:
Comment 6: Please move the following lines to the discussion section (lines 182-183 starting from suggesting…….., 185-186 starting from indicating……….., 188-189 starting from emphasizing………., and 191-192 starting from suggesting)
Response 6: We acknowledge your guidance, and we have made the necessary adjustments in the revised manuscript. Please see the changes in the revised manuscript.
Comment 7: Line 186… is the dose-dependent relationship direct or inverse? Please specify (see Fig 1b). Secondly, do the FS5 (5g/kg) and FS10 (10g/kg) outcomes in Fig 1b, really substantiate a dose-dependent relationship?
Response 7: Thank you for your comment on line 186. We apologize for the confusion. The dose-dependent relationship is neither direct nor inverse, but rather quadratic. We have revised Line 186 and removed the phrase “indicating a dose-dependent relationship between fenugreek seed inclusion and down-regulation of IL8L2.”
Comment 8: Revise the statement in Line 205…. replace it was observed that with “a uniform and significant downregulation in their expression was observed……………”
Response 8: We appreciate your attention to detail and your constructive feedback on our manuscript. We have replaced the original statement to say, “Examining the impact on ileal avian β-defensins (AvBD9, AvBD10, AvBD11), a uniform and significant downregulation in their expression was observed across all levels of fenugreek seed inclusion compared to the control group (Figure 2C, 2D, 2E).” Please see lines 224-227.
Comment 9: Line 292: remove “and” from suggesting and…..
Response 9: Thank you for your comment and suggestion to remove “and” from line 292. Please see line 319.
- Discussion:
Comment 10: Revise lines 301-203 starting with the gut microbiota…
Response 10: Thank you for your comment about starting with the gut microbiota in lines 301-203. We have revised it to start with “Gut microbiota…”Lines 330-332.
Comment 11: Line 311 Avian Leukocytes cytokines: Is the production of cytokines exclusive to leukocytes among different cell types? …how about epithelial cells? Especially when you're measuring the gene expression in the ileum.
Response 11: Thank you for your comment. We agree that cytokines are not exclusive to leukocytes and that epithelial cells can also produce and respond to cytokines. Epithelial cells play an important role in the gut immune homeostasis and inflammation by modulating cytokine signaling and crosstalk with immune cells.
We have revised the sentence to say, “Epithelial cells respond to injury or infection by secreting cytokines, which are regulatory signals that control epithelial cell proliferation, differentiation, and function during inflammation” and updated the citation. Please see lines 343-345.
Comment 12: Line 309… remove ‘s’ from ‘contribute’.
Response 12: Thank you for your suggestion. We have revised this sentence.
Comment 13: Lines 318, 328, 374: italicize Salmonella
Response 13: Thank you for your suggestion. We have italicized the word Salmonella. Please see throughout the manuscript.
Comment 14: Lines 329-335: is the downregulation of AvBD good for the host? It is well-known that AVBD has broad-spectrum antimicrobial activity against various Gram-negative bacteria and enhancing their expression is important for fighting infections. Also, does the downregulation of IRF-7 help the host fight viral infection? The observed downregulation of these genes requires clarification.
Response 14: Thank you for your insightful comment. We agree that AvBDs have broad-spectrum antimicrobial activity and that IRF-7 is important for antiviral responses. However, we also think that the downregulation of these genes in our study may reflect a negative feedback mechanism to prevent excessive inflammation and tissue damage in the intestine as seen in our previous study with the higher inclusion of 5 g and 10 g/ kg fenugreek seeds inclusion (Paneru et al., 2022). Moreover, IRF-7 can activate the expression of type I and III interferons, which can also trigger inflammatory pathways and cause immunopathology. Therefore, we speculate that the downregulation of AvBDs and IRF-7 in our infected chickens may be a protective response to limit the inflammatory damage and restore the homeostasis.
Paneru, D., G. Tellez-Isaias, N. Romano, G. Lohakare, W. G. Bottje, and J. Lohakare. 2022. Effect of Graded Levels of Fenugreek (Trigonella foenum-graecum L.) Seeds on the Growth Performance, Hematological Parameters, and Intestinal Histomorphology of Broiler Chickens. Vet. Sci. 9:207.
Comment 15: Revise statement 346-348: the impact of fenugreek on the host immune system has not been examined in infected or heat-stressed birds in this study.
Response 15: Thank you for your comment. We agree that the statement on lines 346-348 needs to be revised. We have revised the statement to say, “However, further studies are needed to evaluate the impact of FS on the host immune system under different stress conditions, such as infection or heat stress.” Lines 381-383.
Comment 16: The statement in lines 357-358 is awkward.
Response 16: Thank you for your comment. We agree that the statement on lines 357-358 was unclear. We have removed it from the revised manuscript.
Comment 17: Line 384: “In this investigation, the effects of fenugreek seeds on immune-related gene and cecal bacterial diversity were barely apparent” Is this true? How about the significant downregulation of the immune related genes.
Response 18: Thank you for your comment. We acknowledge that the sentence in line 384 was a mistake. We have revised it to say, “In this study, fenugreek seeds had significant impacts on the immune system and the gut microbiota of the chickens.” Lines 419-421.
Comment 19: Segment 392-407, the authors' intended emphasis in this segment is unclear.
Response 19: We appreciate your comment. We intended to highlight the importance of gut microbiota in regulating the immune response in our study in this segment. However, upon careful consideration of your comment, we agree that this segment is not essential for our discussion. Therefore, we have deleted it from the revised manuscript.
- Conclusion:
Comment 20: Lines 410-411 and 413-414 are similar. The conclusion needs to be written to better reflect the outcomes of the study. Future directions should also be included.
Response 20: Thank you for your careful observation. We have revised the conclusion to say, “In conclusion, this study suggests that fenugreek seeds may modulate the expression of immune-related genes and influence the diversity of cecal microbiota in broiler chickens. However, further research is necessary to establish a causal relationship between fenugreek seed consumption and these observed effects. Additionally, future studies should investigate the potential benefits of fenugreek seeds in mitigating the impact of specific poultry pathogens, such as Coccidia or Salmonella, using controlled infection models.”

Reviewer 2 Report
Comments and Suggestions for Authors
1. In the Introduction section, please divide this section into several paragraphs to explain the status quo and hypothesis of this study;
2. In the Materials, Line 88, "Briefly, day-old, male broiler chicks", please revise this sentence, please explain why use male broiler not female in this section;
3. Lines 94-95, please explain briefly why select this dosage of fenugreek seeds for this study;
4. Table1, please provide the sequence for the internal reference beta-actin in the table;
5. Figure 6, please show the results at genus level;
6. Please do correlation analysis to show the relation of cecal microbiota and ileum inflammatory cytokines;
7. Line 300, "gut flora", please revise to "gut microbiota";
8. Some full names for abbreviations appeared for many times in the manuscript, e.g., "PTGS2", "IRF2" lines 308-309, 337-343.
Comments on the Quality of English LanguageEnglish writing is okay for me.
Author Response
Response to Reviewer 2 Comments
|
|
We are grateful for your valuable comments and suggestions on our manuscript. We have carefully revised our manuscript following your comments and have made the corrections. We have also provided a point-by-point response to each comment in the following document. We have addressed all the suggestions in the revised manuscript. Authors’ responses are in red. We hope that our revised manuscript meets your expectations and is now suitable for publication.
|
|
Comment 1: In the Introduction section, please divide this section into several paragraphs to explain the status quo and hypothesis of this study;
Response 1: Thank you for your comment. We have revised the introduction section to divide it into three paragraphs. Please see the introduction.
Comment 2: In the Materials, Line 88, "Briefly, day-old, male broiler chicks", please revise this sentence, please explain why use male broiler not female in this section;
Response 2: Thank you for your comment. We used only male broiler chicks, because they are more commonly raised in commercial settings. Moreover, we did not want to have another variable of sex in our experimental results. Therefore, using male broilers represents a more practical application of our research. We wrote this section briefly because it has already been explained in our previously published article.
Paneru, D., G. Tellez-Isaias, N. Romano, G. Lohakare, W. G. Bottje, and J. Lohakare. 2022. Effect of Graded Levels of Fenugreek (Trigonella foenum-graecum L.) Seeds on the Growth Performance, Hematological Parameters, and Intestinal Histomorphology of Broiler Chickens. Vet. Sci. 9:207.
Comment 3: Lines 94-95, please explain briefly why select this dosage of fenugreek seeds for this study;
Response 3: Thank you for your question. We selected the dosage of fenugreek seeds for this study based on our previous findings and the literature review. Please see Lines 106-107.
Comment 4: Table1, please provide the sequence for the internal reference beta-actin in the table;
Response 4: Thank you for your comment: The primer sequence for internal reference, beta-actin, has already been provided in the last row of Table 1.
Comment 5: Figure 6, please show the results at genus level;
Response 5: Thank you for your comment. We apologize for the inconvenience, but we cannot provide the results at the genus level for Figure 6. The reason for this is that displaying the genus level in Figure 6 will enlarge the figure layout, which could not be used in one figure and the current figure presented the bacterial phyla content in the cecum of broilers. Although we have already added the microbial diversity at the genus level for different treatment groups in Figure 3B. We kindly ask and request to maintain Figure 6 in its present form.
Comment 6: Please do correlation analysis to show the relation of cecal microbiota and ileum inflammatory cytokines;
Response 6: Thank you for your suggestions. We agree with your suggestion, but to do so will increase the number of tables and figures exceeding the journal's guidelines. Instead, we have explained the relationship of cecal microbiota and ileum inflammatory cytokines in the discussion section.
Comment 7: Line 300, "gut flora", please revise to "gut microbiota";
Response 7: Thank you for your suggestions. We have revised the line 327 to say “gut microbiota.”
Rsponse 8: Some full names for abbreviations appeared for many times in the manuscript, e.g., "PTGS2", "IRF2" lines 308-309, 337-343.
Response 8: Thank you for your comment. We have revised Lines 337-338, and 371-373 to include abbreviations for the genes already mentioned in the manuscript.
